# Generalized Encouragement-Based Instrumental Variables for Counterfactual Regression

## Abstract

In causal inference, encouragement designs (EDs) are widely used to analyze causal effects, when randomized controlled trials (RCTs) are impractical or compliance to treatment cannot be perfectly enforced. Unlike RCTs, which directly allocate treatments, EDs randomly assign encouragement policies that positively motivate individuals to engage in a specific treatment. These random encouragements act as instrumental variables (IVs), facilitating the identification of causal effects through leveraging exogenous perturbations in discrete treatment scenarios. However, real-world implementations of EDs often deviate from ideal conditions. Specifically, encouragements are frequently nonrandom, the number of available encouragement policies is limited, and sample sizes are often small—posing significant challenges to reliable causal estimation. To address this, we propose a novel identifiability theory that leverages variations in encouragement to identify the Conditional Average Treatment Effect (CATE). Building on this foundation, we develop a new IV estimator, named **En**couragement-based **Counter**factual **R**egression (**EnCounteR**), to effectively estimate causal effects even when the number of instruments is smaller than the number of treatments. Extensive experiments on both synthetic and real-world datasets demonstrate the superiority of the proposed EnCounteR.

## 1 Introduction

Causal inference is a powerful tool for explanatory analysis and plays a crucial role in fields such as healthcare, economics, and social sciences (Imbens et al., 2015; Wooldridge et al., 2016; Burgess et al., 2017; Devriendt et al., 2020; Kuang et al., 2020). Although Randomized Controlled Trials (RCTs) are the gold standard for identifying causal relationships under unmeasured confounders, they often face issues such as noncompliance and ethical concerns (Kohavi & Longbotham, 2011; Bottou et al., 2013; Wu et al., 2022b). As an alternative, randomized encouragement designs (EDs), which randomly assign encouragement policies to positively influence individuals' likelihood of receiving a treatment, have gained popularity (Small et al., 2008; West et al., 2008) and are increasingly used to estimate causal effects in practice (Holland, 1988; Small, 2007; Kang & Imbens, 2016). For example, Sexton & Hebel (1984) and Permutt & Hebel (1989) encouraged physicians to advise against smoking to study its impact on birth weight, while Angrist et al. (1996), Bang & Davis (2007), and Kang & Imbens (2016) employed randomized intent-to-treat designs to address treatment non-compliance.

In the above encouragement designs, random encouragements serve as instrumental variables (IVs), positively influencing treatment uptake without directly affecting the outcome, thereby satisfying the relevance, exclusion restriction, and exogeneity assumptions (Angrist & Imbens, 1995; Angrist et al., 1996; Hartford et al., 2017). However, such discrete encouragement-based IVs are typically limited to identifying the local average treatment effect (LATE) in discrete treatment settings under the monotonicity assumption (Angrist et al., 1996; Pearl, 2010; Wooldridge et al., 2016). When the number of encouragements is substantially smaller than the number of treatment options — especially in the case of continuous treatments — the induced variation in treatment tends to be weak and sparse.

Such settings, where treatment is continuous while encouragement has multiple but limited values, are ubiquitous and prevalent in many real-world applications. For example, as shown in Figure 1, on online course platforms such as Coursera, edX, and Udacity (Breslow et al., 2013; Reich, 2015; Anderson et al., 2014; Kizilcec et al., 2014), instructors apply various encouragements (e.g., $e_A$ = None, $e_B$ = Praise, $e_C$ = Points) to motivate students toward longer forum participation

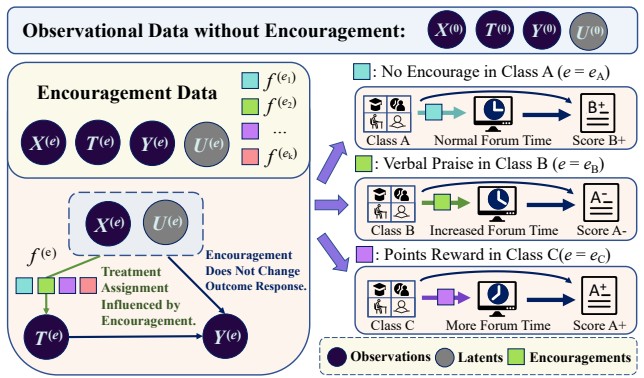

Figure 1: Framework overview.

(i.e., treatments $T$), while these encouragements do not have a direct impact on exam scores $Y$, which offers opportunities to identify causal effects in the presence of unmeasured confounders $U$. In this case, the type of encouragement is limited while the forum participation time is continuous. Such insufficient and unevenly distributed treatment variation poses significant challenges in identifying causal effects, violating the condition that the support set of IVs must exceed that of treatments. Moreover, in real-world applications, instructors often select encouragements based on class characteristics, resulting in nonrandom encouragement assignments. This introduces endogeneity issues and further exacerbates the challenge of accurately identifying causal effects.

In this paper, we aim to address the challenges inherent in settings with limited encouragement and continuous treatment, including: (1) limited experimental data, (2) non-random encouragement assignments, and (3) a small number of encouragement strategies. First of all, to improve sample efficiency, we treat the large observational data $\{X^{(0)}, T^{(0)}, Y^{(0)}\}$ as arising from a special encouragement condition $e = e_0$, and leverage it as auxiliary information to identify the CATE through small-scale encouragement experiments. We then apply covariate shift balancing, a technique that simulates random encouragement assignments and mitigates endogeneity issues caused by non-random encouragement. Based on this encouragement data, we propose novel identifiability theory and develop a new IV estimator, named **En**couragement-based **Counte**rfactual **R**egression (**EnCounteR**), to estimate causal effects. Notably, conventional two-stage IV regression is a special case of our EnCounteR. Empirical evaluations demonstrate the superiority of EnCounteR.

## 2 RELATED WORK

**Encouragement designs** have been widely used to analyze causal effects, when RCTs are impractical or compliance and treatment cannot be perfectly enforced (Sexton & Hebel, 1984; Permutt & Hebel, 1989; Angrist et al., 1996; Bang & Davis, 2007; Zhang et al., 2022). Angrist et al. (1996), Hirano et al. (2000), Bang & Davis (2007), and Kang & Imbens (2016) employed random intent-to-treat as instruments to encourage treatment for addressing non-compliance issues. Fletcher (2010), An (2015), and Kang & Imbens (2016) utilized personalized encouragement assumptions to study the peer effect in school settings. However, real applications of encouragement designs often pose challenges, including non-randomized encouragements, limited experimental data, and a smaller number of encouragements compared to continuous treatments, hindering precise causal effect estimation.

**Instrumental variables** induce exogenous perturbations to the treatment variable, allowing for the estimation of causal effects in the presence of unmeasured confounders (Hirano et al., 2000; Sovey & Green, 2011; An, 2015; Cheng et al., 2023; Sun et al., 2024; Zhao et al., 2024). Traditional IV two-stage regression first identifies treatment variation caused by IVs and then uses it to estimate the dependent variable (Wald, 1940; Angrist & Imbens, 1995; Angrist et al., 1996). Based on sieve theory (Newey & Powell, 2003), researchers have developed numerous non-linear IV variants (Hartford et al., 2017; Singh et al., 2019; Muandet et al., 2020; Bennett et al., 2019; Dikkala et al., 2020; Xu et al., 2021; Wu et al., 2022a). However, in continuous treatments with limited IVs, the identification condition of IVs, requiring that the support set of IVs be larger than that of treatments, can be violated.

**Multiple Environments**: Invariant learning across multiple environments (Arjovsky et al., 2019; Duchi & Namkoong, 2021; Creager et al., 2021; Liu et al., 2021a;b; Wang et al., 2023) has been studied. Arjovsky et al. (2019) identified causally invariant relationships in different environments, assuming their existence for exploration. Liu et al. (2021a;b) generated environments and proposed a maximal invariant predictor, integrating environment inference with invariant learning. However, these works can only estimate the total effect, rather than the pure causal effect of treatments.

## 3 PROBLEM SETUP AND SOLUTIONS

### 3.1 NOTATIONS

Following Liu et al. (2021a;b), we consider a dataset $\mathcal{D} = \{\mathcal{D}^{(e_k)}\}_{e_k \in \mathcal{E}}$, which comprises multiple datasets $\mathcal{D}^{(e_k)} = \{x_i^{(e_k)}, t_i^{(e_k)}, y_i^{(e_k)} \mid u_i^{(e_k)}\}_{i=1}^{n_k}$ under different encouragement designs $e_k$ in $\mathcal{E} = \{e_0, e_1, \cdots, e_K\}$, and $n_k$ is sample size in encouragement $e_k$. Within each dataset $\mathcal{D}^{(e_k)}$, the variables $x_i^{(e_k)} \in \mathcal{X}$ and $u_i^{(e_k)} \in \mathcal{U}$ are respectively the observable and unmeasured confounders, potentially confounding the analysis of the causal effect of the treatment variables $t_i^{(e_k)} \in \mathcal{T}$ on the outcome variables $y_i^{(e_k)} \in \mathcal{Y}$. As illustrated in Figure 1, observational data alone cannot identify the Conditional Average Treatment Effects (CATE) due to unmeasured confounders. Therefore, we apply $K$ different encouragement policies to promote treatment adoption without directly manipulating the treatment in certain candidate groups $\mathcal{D}^{(e_k)} = \{x_i^{(e_k)}, t_i^{(e_k)}, y_i^{(e_k)} \mid u_i^{(e_k)}\}_{i=1}^{n_k}$:

$$t_i^{(e_k)} = f_\Phi^{(e_k)}(x_i^{(e_k)}, u_i^{(e_k)}), \quad y_i^{(e_k)} = g_\Psi(t_i^{(e_k)}, x_i^{(e_k)}) + \varepsilon(u_i^{(e_k)}), \tag{1}$$

where $f_\Phi^{(e_k)}(\cdot)$ denotes different treatment assignment mechanisms with unknown parameters $\Phi^{(e_k)}$ for various encouragements $e_k \in \mathcal{E}$, $g_\Psi(\cdot)$ is the heterogeneous treatment effect with unknown parameters $\Psi$, and $\varepsilon(\cdot)$ embeds the unmeasured confounding effects from $u_i^{(e_k)}$, additive noise that is commonly assumed in causality (Newey & Powell, 2003; Imbens et al., 2015; Hartford et al., 2017).

Traditional approaches assume that encouragements are randomly assigned and exogenous, i.e., $U \perp\!\!\!\perp \mathcal{E}$, to ensure valid causal estimation. However, in real-world applications, practitioners often assign encouragements non-randomly to pre-existing groups—such as classes or cities—introducing potential endogeneity. For example, if encouragement is given in classes led by more experienced teachers, it becomes difficult to discern whether improved student outcomes stem from the encouragement or the teacher's quality. To mitigate this issue, we collect rich covariates $X$ as proxies for $U$ and propose a covariate balancing module that addresses distribution shifts induced by non-random encouragements. This allows us to relax the independence assumption to conditional independence.

To simplify notation, we denote random variables as uppercase notation, $X = \{X^{(e_k)}\}_{e \in \mathcal{E}}$, where $X^{(e_k)} = \{x_i^{(e_k)}\}_{i=1}^{n^{(e_k)}}$ signifies the sample vector of observed pre-treatment variables for each encouragement design $e_k$. Similarly, we define $U^{(e_k)}, T^{(e_k)}$, and $Y^{(e_k)}$ as the corresponding vectors under each $e_k$. We use $\mathbb{E}[X^{(e_k)}]$, $\mathrm{Var}(X^{(e_k)})$, and $\mathrm{Cov}(Y^{(e_k)}, X^{(e_k)})$ to denote the expectation, variance, and covariance between $Y^{(e_k)}$ and $X^{(e_k)}$, respectively. We then define $\beta_{Y|X}^{(e_k)} = \frac{\mathrm{Cov}(Y^{(e_k)}, X^{(e_k)})}{\mathrm{Var}(X^{(e_k)})}$.

### 3.2 ASSUMPTIONS AND THEOREMS

In many real applications, encouragement designs suffer from nonrandom encouragements, limited samples, and sparse encouragement policies, resulting in unreliable causal estimations. To address this, we leverage both observational and encouragement data $\mathcal{D} = \{\mathcal{D}^{(e_k)}\}_{e_k \in \mathcal{E}}$ and develop novel theory and algorithms to identify causal effects on outcomes. To this end, we naturally start with a linear setting to build intuition on the necessary assumptions and corresponding theorems, and then put efforts into generalizing these insights to more complex nonlinear settings.

#### 3.2.1 FORMALIZATION IN LINEAR SETTING

For illustration, consider a linear reformulation of Eq. 1:

$$y_i^{(e_k)} = \psi_t t_i^{(e_k)} + \psi_x x_i^{(e_k)} + \psi_u u_i^{(e_k)}. \tag{2}$$

where the coefficient $\psi_t$ is the constant causal effect of interest, and the treatment assignments $t_i^{(e_k)} = f_\Phi^{(e_k)}(x_i^{(e_k)}, u_i^{(e_k)})$ can be arbitrary functions across encouragements $e_k$. Under Assumption 1 and 2, we propose a novel identification theorem of causal effect $\psi_t$.

**Assumption 1** (Linearity). *The outcome variable $Y$ is a linear function of variables $T$, $X$, and $U$.*

**Assumption 2** (Independence). *$X$ and $U$ are independent of the encouragements, i.e., $\{X, U\} \perp\!\!\!\perp \mathcal{E}$.*

This assumption arises from the common linear case. In subsequent non-linear settings, we will relax the full independence assumption and retainretain only $U \perp\!\!\!\perp \mathcal{E} \mid X$ for greater flexibility.

**Theorem 1.** *Under Assumptions 1 & 2, given datasets $\{\mathcal{D}^{(e_0)}, \mathcal{D}^{(e_1)}, \cdots, , \mathcal{D}^{(e_K)}\}$ with different encouragements $\{e_0, e_1, \cdots, e_K\} \in \mathcal{E}$, the causal effect $\psi_t$ is identified by $\psi_t = \frac{\beta_{Y|X}^{(e_1)} - \beta_{Y|X}^{(e_0)}}{\beta_{T|X}^{(e_1)} - \beta_{T|X}^{(e_0)}}$, regardless of whether the treatment $T$ is binary, multi-valued, or continuous.*

The proofs are deferred to Appendix A. Traditional IV regression can be seen as a special case of our framework, where $X$ serves as the instrument for $T$ on $Y$. For example, in observational IV data $\mathcal{D}^{(e_0)}$, if $X$ are instrumental variables that are independent of unmeasured confounders $U$ and have no direct effect on $Y$, then the causal effect is given by $\psi_t = \beta_{Y|X}^{(e_0)} / \beta_{T|X}^{(e_0)}$.

**Corollary 1.** *With $\psi_x = 0$ and $X \perp\!\!\!\perp U$ in linear model Eq. 2, given the observations $\mathcal{D}^{(e_0)}$, then $\psi_t$ is identifiable by $\psi_t = Cov(Y, X) / Cov(T, X) = \beta_{Y|X}^{(e_0)} / \beta_{T|X}^{(e_0)}$.*

**Theorem Contribution:** The existing IV literature (Angrist & Imbens, 1995; Cameron & Trivedi, 2005; Wooldridge, 2010; Hernan & Robins, 2010; Hayashi, 2011; Wooldridge, 2015; Hansen, 2022) has primarily focused on addressing unobserved confounding. In these frameworks, observed covariates are often either ignored or implicitly absorbed into the treatment variable, resulting in identification theorems that typically model $\{Z, X\} \to \{T, X\} \to Y$. Consequently, most approaches implicitly assume that the number of valid instruments must be at least as large as the number of treatment variables to ensure identification (Angrist & Krueger, 2001; Cameron & Trivedi, 2005; Dougherty, 2011; Wooldridge, 2010; 2015). To the best of our knowledge, we are the first to propose a theoretical framework that accommodates binary or discrete instrumental variables in continuous treatment cases, while actively leveraging heterogeneity in observed covariates $X$ across different IV strata to facilitate identification. This novel approach allows for the identification of causal effects even when the number of instruments is smaller than the number of treatment variables.

### 3.2.2 GMM Reformulation

Theorem 1 provides a linear analytical solution (LAS) for encouragement datasets. When the number of encouragements exceeds two, the system becomes over-identified, as it contains more equations than unknowns. To resolve this, we reformulate the issue as a generalized method of moments (GMM) problem. We use a residual $\epsilon_i^{(e_k)}$ to identify the parameters $\{\psi_t, b = \psi_x + \psi_u \beta_{U|X}^{(e_k)}\}$, where the residual is defined as: $\epsilon_i^{(e_k)} = y_i^{(e_k)} - \psi_t t_i^{(e_k)} - b x_i^{(e_k)} = \psi_u(u_i^{(e_k)} - \beta_{U|X}^{(e_k)} x_i^{(e_k)})$. Here, $\epsilon_i^{(e_k)}$ is not simply $u_i^{(e_k)}$ but a transformed residual designed to remove its correlation with $x_i^{(e_k)}$.

**Theorem 2.** *Under Linearity Assumption 1, $\frac{1}{\psi_u} \text{Cov}(\epsilon^{(e_k)}, X^{(e_k)}) = \text{Cov}(U^{(e_k)}, X^{(e_k)}) - \frac{\text{Cov}(U^{(e_k)}, X^{(e_k)})}{\text{Var}(X^{(e_k)})} \text{Var}(X^{(e_k)})$. Accordingly, $\text{Cov}(\epsilon^{(e_k)}, X^{(e_k)}) = 0$ for any $e_k \in \mathcal{E}$.*

Define $\tilde{\epsilon}^{(e_k)} = \epsilon^{(e_k)} - \mathbb{E}[\epsilon^{(e_k)}] = \tilde{Y}^{(e_k)} - \psi_t \tilde{T}^{(e_k)} - b\tilde{X}^{(e_k)}$, where $\{\tilde{T}, \tilde{X}, \tilde{Y}\}$ are de-meaned variables. Based on Theorem 2, i.e., $\mathbb{E}[\tilde{\epsilon}^{(e_k)} \tilde{X}^{(e_k)}] = \text{Cov}(\epsilon^{(e_k)}, X^{(e_k)}) = 0$, we can derive $K + 1$ moments for $\mathcal{E} = \{e_0, e_1, \cdots, e_K\}$:

$$g_X(\psi_t, b) = \begin{bmatrix} \mathbb{E}[(\tilde{Y}^{(e_0)} - \psi_t \tilde{T}^{(e_0)} - b\tilde{X}^{(e_0)})\tilde{X}^{(e_0)}] \\ \cdots \\ \mathbb{E}[(\tilde{Y}^{(e_K)} - \psi_t \tilde{T}^{(e_K)} - b\tilde{X}^{(e_K)})\tilde{X}^{(e_K)}] \end{bmatrix}. \quad (3)$$

Since the function $Y^{(e_k)} - \hat{Y}_{\psi_t, b}^{(e_k)}$ is only related to $U$ and $X$, where $\hat{Y}_{\psi_t, b}^{(e_k)} = \hat{\psi}_t T^{(e_k)} + \hat{b} X^{(e_k)}$, under Independence Assumption 2, we can conclude that $[Y^{(e_k)} - \hat{Y}_{\psi_t, b}^{(e_k)}] \perp\!\!\!\perp \mathcal{E}$:

$$g_{\mathcal{E}}(\psi_t, b) = \begin{bmatrix} \mathbb{E}[Y^{(e_i)} - \hat{Y}_{\psi_t, b}^{(e_i)}] - \mathbb{E}[Y^{(e_j)} - \hat{Y}_{\psi_t, b}^{(e_j)}] \\ \text{Var}[Y^{(e_i)} - \hat{Y}_{\psi_t, b}^{(e_i)}] - \text{Var}[Y^{(e_j)} - \hat{Y}_{\psi_t, b}^{(e_j)}] \end{bmatrix}_{i \neq j}, \quad (4)$$

and the GMM estimator can be written as follows with non-negative definite matrices $\{W_X, W_{\mathcal{E}}\}$:

$$(\psi_t^*, b^*) = \arg \min_{\hat{\psi}_t, \hat{b}} [g_X' \cdot W_X \cdot g_X + g_{\mathcal{E}}' \cdot W_{\mathcal{E}} \cdot g_{\mathcal{E}}]. \quad (5)$$

The optimal $W^*$ depends on the moments covariance matrix.

### 3.2.3 GENERALIZATION TO NON-LINEAR SETTINGS

Recall the generalized non-linear settings in Eq. 1, i.e., $y_i^{(e_k)} = g_\Psi(t_i^{(e_k)}, x_i^{(e_k)}) + \varepsilon(u_i^{(e_k)})$, where the outcome response function $g_\Psi(\cdot)$ and noise $\varepsilon(\cdot)$ remain constant across different encouragements. Under the classical IV assumptions (Newey & Powell, 2003; Hartford et al., 2017), one can first identify the transformed outcome:

$$h_\theta(T, X) = g_\Psi(T, X) + \mathbb{E}[\varepsilon(U) \mid X]. \tag{6}$$

**Assumption 3** (Encouragement-Based Instrumental Variable). *The adopted encouragement policies $e \in \mathcal{E}$ serve as IVs, which only positively motivate the choice of treatments, without directly affecting the outcome response, which satisfies the following three IV conditions:*
*(a)* Relevance*: IVs would directly affect $T^e$, i.e., $T^e \not\perp\!\!\!\perp e$;*
*(b)* Exclusion*: IVs do not directly affect $Y^e$, i.e., $Y^{e_i}(t, x) = Y^{e_j}(t, x)$ for $e_i \neq e_j$ and all $t$ and $x$;*
*(c)* Exogeneity*: IVs are conditional independent of the error $\varepsilon(U)$, i.e., $e \perp\!\!\!\perp \varepsilon(U) \mid X$.*

**Assumption 4** (Additive Noise). *$\varepsilon(u_i^{(e_k)})$ embeds the confounding effect from $u_i^{(e_k)}$ as an additive noise term with $\mathbb{E}[\varepsilon(U^{(e_k)})] = 0$ (Hartford et al., 2017).*

Under Assumptions 3 and 4, the encouragements could be seen as valid IVs, Then the CATE is identified by $\text{CATE}(T, X) = h_\theta(T, X) - h_\theta(0, X)$.

**Theorem 3.** *When the relevance between $T$ and $e$ is strong, the unique solution $h_\theta(T, X)$ is identified by the inverse problem given $\mathbb{E}[Y \mid e, T, X]$ and conditional treatment distribution $dF(T \mid e, X)$:*

$$\mathbb{E}[Y \mid e, T, X] = \int [h_\theta(T, X)] \, dF(T \mid e, X). \tag{7}$$

*Its proof can be found in (Newey & Powell, 2003).*

However, discrete encouragements $e$ may only introduce a minor exogenous disturbance to the continuous treatment that is too small to accurately estimate CATE. To address this issue, we propose a novel discrete encouragement algorithm by combining Theorems 1 and 3 to extend the moment conditions in Eq. 5 to a non-linear setting:

$$g_R(\theta, \xi) = \begin{bmatrix} \mathbb{E}[(Y^{(e_0)} - h_\theta(T^{(e_0)}, X^{(e_0)}))r_\xi(X^{(e_0)})] \\ \cdots \\ \mathbb{E}[(Y^{(e_K)} - h_\theta(T^{(e_K)}, X^{(e_K)}))r_\xi(X^{(e_K)})] \end{bmatrix}. \tag{8}$$

$$g_\mathcal{E}(\theta) = \begin{bmatrix} \mathbb{E}[Y^{(e_i)} - \hat{Y}_\theta^{(e_i)}] + \mathbb{E}[Y^{(e_j)} - \hat{Y}_\theta^{(e_j)}] \\ \mathbb{E}[Y^{(e_i)} - \hat{Y}_\theta^{(e_i)}] - \mathbb{E}[Y^{(e_j)} - \hat{Y}_\theta^{(e_j)}] \\ \text{Var}[Y^{(e_i)} - \hat{Y}_\theta^{(e_i)}] - \text{Var}[Y^{(e_j)} - \hat{Y}_\theta^{(e_j)}] \end{bmatrix}_{i \neq j}. \tag{9}$$

where $\hat{Y}_\theta = h_\theta(T, X)$ and $r_\xi(\cdot)$ is the representations of $X$, providing non-linear moments. Eq. 9 ensures the expectation of residual is zero and independent of $\mathcal{E}$.

$$(\theta^*, \xi^*) = \arg\min_{\hat{\theta}} \sup_{\hat{\xi}} \left[ l(Y, \hat{Y}_\theta) + g_R' W_R g_R + g_\mathcal{E}' W_\mathcal{E} g_\mathcal{E} \right], \tag{10}$$

where $l(\cdot)$ represents the cross-entropy loss for binary outcomes or mean squared error for continuous outcomes, while the moments constraints $g_R' W_R g_R$ and $g_\mathcal{E}' W_\mathcal{E} g_\mathcal{E}$ act as penalties aiding in the estimation of $\hat{Y}_\theta$, where $\{W_R, W_\mathcal{E}\}$ are non-negative definite weighting matrices, with the optimal $W^*$ determined by the moments covariance matrix.

**Corollary 2.** *Under Assumptions 3 & 4, the result of estimated $\theta^*$ in Eq. 10 equals $h_\theta(T, X)$.*

Our algorithm differs from DeepGMM (Bennett et al., 2019) and AGMM (Dikkala et al., 2020) that require $X$ to be exogenous. This condition is hard to satisfy in reality. Besides, these methods ignore keeping the residual expectation to be zero while minimizing the regression error. To this end, under nonrandom encouragements and continuous treatments, we develop novel theory and algorithms for identifying and estimating CATE. When the covariates $X^{(e)}$ shift slightly across encouragements, we reweight samples to estimate causal effects, with the Exogeneity Assumption 3(c).

## 4 METHODOLOGY

Combining observational and encouragement data in $\mathcal{D} = \{\mathcal{D}^{(e_k)}\}_{e_k \in \mathcal{E}}$, we follow the theoretical insights from the previous sections to train neural networks $h_\theta$ with moment constraints for

Encouragement-based Counterfactual Regression (EnCounteR). Specifically, our model's overall architecture comprises the following components: (1) Covariate balancing under Exogeneity Assumption 3(c); (2) full moment constraints with adversarial representation matrices; (3) counterfactual regression with moment constraints. Next, we will introduce each module step by step.

## 4.1 COVARIATE SHIFT BALANCING

As depicted in Figure 1, we collect large observational data $\mathcal{D}^{(e_0)}$ from previous samples and implement $K$ encouragements $\{\mathcal{D}^{(e_k)}\}_{1 \leq k \leq K}$ in new samples to examine the causal effect of $T$ on $Y$. However, samples under different encouragements may exhibit slight covariate shifts, such as minor differences between two different classes in the same school. Therefore, we introduce the following Reweighting module to balance observed covariates across various environments:

$$\mathcal{L}_\omega = \sum_{j \neq k} (\mathbb{E}_\omega X^{(e_j)} - \mathbb{E}_\omega X^{(e_k)})^2 + (\text{Cov}_\omega X^{(e_j)} - \text{Cov}_\omega X^{(e_k)})^2,$$
$$\mathbb{E}_\omega X^{(e_k)} = \omega^{(e_k)\prime} X^{(e_k)}, \text{Cov}_\omega X^{(e_k)} = \tilde{X}^{(e_k)\prime} \omega \tilde{X}^{(e_k)}, \quad (11)$$
$$\omega^{(e_k)} = [(1 + 3\sigma(w^{(e_k)}))/2]/[\sum_i^{n_j}(1 + 3\sigma(w^{(e_k)}))/2],$$

where $\sigma(\cdot)$ is the sigmoid function, and $w^{(e_k)}$ are trainable parameters with $n_k$ units. The term $\frac{1+3\sigma(w)}{2} \in [\frac{1}{2}, 2]$ serves to limit extreme values during the reweighting process.

## 4.2 MOMENT CONSTRAINT LEARNING

Following the weight $\omega$ from Eq. 11, we define $\mathbb{E}_\omega$ as weighted expectation and $\text{Var}_\omega$ as weighted variance, and then construct moment constraints to learn $h_\theta(T, X)$ and $r_\xi(X)$.

(I) Encouragement-Independent Moments:

$$\mathcal{L}_\mathcal{E} = g'_\mathcal{E}(\theta) \cdot W_\mathcal{E} \cdot g_\mathcal{E}(\theta), \quad g_\mathcal{E}(\theta) = \begin{bmatrix} \mathbb{E}_\omega[Y^{(e_i)} - \hat{Y}_\theta^{(e_i)}] + \mathbb{E}_\omega[Y^{(e_j)} - \hat{Y}_\theta^{(e_j)}] \\ \mathbb{E}_\omega[Y^{(e_i)} - \hat{Y}_\theta^{(e_i)}] - \mathbb{E}_\omega[Y^{(e_j)} - \hat{Y}_\theta^{(e_j)}] \\ \text{Var}_\omega[Y^{(e_i)} - \hat{Y}_\theta^{(e_i)}] - \text{Var}_\omega[Y^{(e_j)} - \hat{Y}_\theta^{(e_j)}] \end{bmatrix}_{i \neq j}, \quad (12)$$

where $\hat{Y}_\theta = h_\theta(T, X)$ and residual $\epsilon = Y - \hat{Y}_\theta$. Eq. 12 guarantees that $\epsilon \perp\!\!\!\perp e$.

(II) Covariate-Independent Moments:

$$\mathcal{L}_X = g'_X(\theta, \xi) \cdot W_X \cdot g_X(\theta, \xi), \quad g_X(\theta, \xi) = \begin{bmatrix} \mathbb{E}_\omega[(Y^{(e_0)} - h_\theta(T^{(e_0)}, X^{(e_0)}))\tilde{X}^{(e_0)}] \\ \cdots \\ \mathbb{E}_\omega[(Y^{(e_K)} - h_\theta(T^{(e_K)}, X^{(e_K)}))\tilde{X}^{(e_K)}] \end{bmatrix}. \quad (13)$$

Equation 13 ensures that the residual ($\epsilon = Y - \hat{Y}_\theta$) and covariates are linearly independent.

(III) Adversarial Representation-Independent Moments:

$$\mathcal{L}_R = g'_R(\theta, \xi) \cdot W_R \cdot g_R(\theta, \xi), \quad g_R(\theta, \xi) = \begin{bmatrix} \mathbb{E}_\omega[(Y^{(e_0)} - h_\theta(T^{(e_0)}, X^{(e_0)}))r_\xi(X^{(e_0)})] \\ \cdots \\ \mathbb{E}_\omega[(Y^{(e_K)} - h_\theta(T^{(e_K)}, X^{(e_K)}))r_\xi(X^{(e_K)})] \end{bmatrix}. \quad (14)$$

In complex non-linear settings, the underlying independence assumptions typically entail an infinite set of moment conditions. Consequently, we employ Adversarial Representation Learning to learn non-linear factors $R = r_\xi(X) \in \mathbb{R}^{d_r}$ for adaptively constructing the top-$d_r$ moment conditions in minimax criterion (see Eq. 16). In Eqs. (12-14), $W_\mathcal{E}$, $W_X$ and $W_R$ are non-negative definite weighting matrices, and the optimal $W^*$ depends on the moments covariance matrix.

## 4.3 COUNTERFACTUAL REGRESSION

Before proceeding with counterfactual regression $h_\theta(\cdot)$, we conduct a statistical test to check if the mean and covariance of $X$ are independent of encouragements; if not, we perform a preprocessing step to learn $\omega$ for achieving covariate balance, as detailed in Section 4.1. We employ two-layer neural networks with ELU activation, where each layer comprises $d_h$ hidden units for $d_r$-dimensional Representation $R = r_\xi(X)$ and Counterfactual Regression $\hat{Y}_\theta = h_\theta(T, X)$:

$$\mathcal{L}_{\text{REG}} = \mathbb{E}_\omega[l(Y, h_\theta(T, Y))]. \quad (15)$$

Following Theorems 3 and Corollary 2, the complete objective function is formulated as follows:

$$\arg\min_\theta \sup_\xi \mathcal{L} = \mathcal{L}_{\text{REG}} + \alpha(\mathcal{L}_\mathcal{E} + \mathcal{L}_X + \mathcal{L}_R), \quad (16)$$

where $\alpha$ is a hyper-parameter. More implementation details can be found in Appendix B.

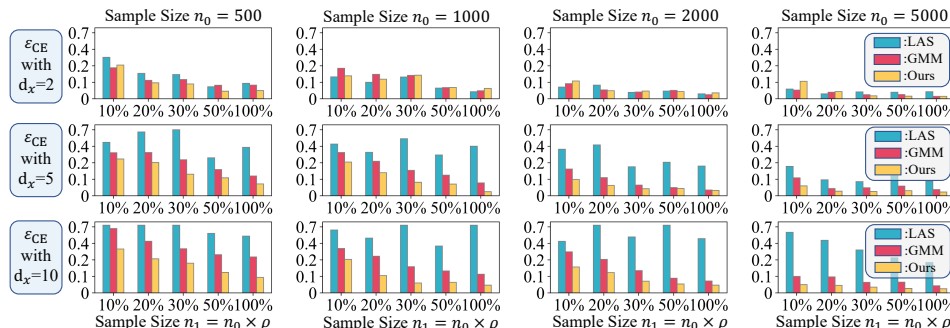

Figure 2: Results ($\varepsilon_{\text{CE}}$) of LAS, GMM, and Our EnCounteR in Linear Simulations, with varying sample sizes $n_0 \in \{500, 1000, 2000, 5000\}$ for observational dataset $\mathcal{D}^{(e_0)}$ and varying sample sizes $n_1 = n_0 \times \rho$ with $\rho = \{10\%, 20\%, 30\%, 50\%, 100\%\}$ for encouragement experiments $\mathcal{D}^{(e_1)}$ across various dimensions $d_x = \{2, 5, 10\}$ of $X$.

## 5 EMPIRACAL EXPERIMENTS

### 5.1 BASELINES AND METRICS

For comparison, we choose baselines from two groups. The first are *instrument-based methods*: (1) **KernelIV** (Singh et al., 2019) and **DualIV** (Muandet et al., 2020) implement kernel ridge regression derived from reproducing kernel Hibert spaces. (2) **DeepGMM** (Bennett et al., 2019) and **AGMM** (Dikkala et al., 2020) construct structural functions and search moment conditions via adversarial training. (3) **DeepIV** (Hartford et al., 2017), **DFIV** (Xu et al., 2021) and **CBIV** (Wu et al., 2022a) run two-stage regression using deep neural networks. The other group is *covariate-based methods*: (1) **CFRNet** (Shalit et al., 2017) and **DRCFR** (Hassanpour & Greiner, 2020) use mutual information to learn balanced representations in continuous cases, while **VCNet** (Nie et al., 2021) is tailored to continuous treatment, preserving the continuity of the estimated counterfactual curve. (2) **CEVAE** (Louizos et al., 2017) and **TEDVAE** (Zhang et al., 2021) employ latent variable modeling to concurrently estimate the latent space summarizing confounders and the causal effects. (3) **KerIRM** and **KerHRM** aim to identify causally invariant relationships in different environments, with the former using known encouragement labels and the latter not using them (Arjovsky et al., 2019; Liu et al., 2021a;b). Additionally, we employ the **VANILLA** network regression as a baseline.

In this section, we use three key metrics for evaluation purposes: $\varepsilon_{\text{CE}} = |\hat{\psi}_t - \psi_t|$ measures causal parameter estimation accuracy in linear simulations; $\varepsilon_{\text{CFR}} = \mathbb{E}(\hat{Y}_\theta(do(t), X) - Y(do(t), X))^2$ assesses the precision of counterfactual outcome predictions using mean square error, where $do(t)$ denotes do operations randomly sampled from a uniform distribution $\text{Unif}[0, 1]$; and the Precision in Estimation of Heterogeneous Effect is measured by $\varepsilon_{\text{PEHE}} = \sqrt{\mathbb{E}(\text{CATE}(do(t), X) - \text{CATE}(do(0), X))^2}$.

### 5.2 EXPERIMENTS ON LINEAR SIMULATIONS

**Datasets** In linearity scenarios, we collect samples $\mathcal{D}^{(e_0)}$ with varying sizes $n_0 \in \{500, 1000, 2000, 5000\}$. Then, we conduct a single encouragement experiment $e_1$ on a smaller dataset $\mathcal{D}^{e_1}$, where we manipulate the experimental data size by setting $n_1 = n_0 \times \rho$ with different proportions $\rho = \{10\%, 20\%, 30\%, 50\%, 100\%\}$ to investigate the impact of sample size on performance of our EnCounteR. Subsequently, we generate a combined dataset $\{\mathcal{D}^{(e_k)}\}_{k \in \{0,1\}} = \{X^{(e_k)}, U^{(e_k)}, T^{(e_k)}, T^{(e_k)}\}_{k \in \{0,1\}}$ with $T^{(e_k)} = \phi_x^{(e_k)\prime} X^{(e_k)} + \phi_u^{(e_k)\prime} U^{(e_k)}$ and $Y^{(e_k)} = \psi_t T^{(e_k)} + \psi_x' X^{(e_k)} + \psi_u' U^{(e_k)}$, where $d_x$-dimensional $X$ and $d_u$-dimensional $U$ are generated from a Normal Distribution $\mathcal{N}(0, 1)$ with common covariance of 0.3, and we set $d_x \in \{2, 5, 10\}$ and $d_u = 2$. The corresponding coefficients $\{\phi_x^{(e_k)}, \phi_u^{(e_k)}, \psi_x, \psi_u\}$ are independently sampled from a Uniform distribution $\text{Unif}(0, 1)$. In our experiments, $\psi_t$ is the causal parameter of interest, and we set it to $\psi_t = 0.5$. We leverage $\{\mathcal{D}^{(e_k)}\}_{k \in \{0,1\}}$ as training data, reserving 10%-30% of $\mathcal{D}^{(e_0)}$ as validation data, and generate 20,000 additional samples with random treatments $do(t) \sim \text{Unif}[0, 1]$ and its corresponding outcome $Y(do(t), X)$ as testing data. We conduct 10 replications for each experiments.

Table 1: Results (mean$_{\pm\text{std}}$) of $\epsilon_{\text{CFE}}$ and $\epsilon_{\text{PEHE}}$ on Simulation, IHDP and ACIC Datasets.

| Methods | Simulation (MULT) | | IHDP | | ACIC | |
|---|---|---|---|---|---|---|
| | $\epsilon_{\text{CFE}}$ | $\epsilon_{\text{PEHE}}$ | $\epsilon_{\text{CFE}}$ | $\epsilon_{\text{PEHE}}$ | $\epsilon_{\text{CFE}}$ | $\epsilon_{\text{PEHE}}$ |
| **KernelIV** | $17.44_{\pm 2.147}$ | $0.611_{\pm 0.153}$ | $3.808_{\pm 1.279}$ | $0.581_{\pm 0.046}$ | $38.82_{\pm 2.457}$ | $0.602_{\pm 0.023}$ |
| **DualIV** | $92.64_{\pm 44.39}$ | $2.454_{\pm 0.679}$ | $19.60_{\pm 4.877}$ | $2.537_{\pm 0.248}$ | $28.41_{\pm 3.384}$ | $0.752_{\pm 0.047}$ |
| **DeepGMM** | $6.340_{\pm 2.177}$ | $0.584_{\pm 0.105}$ | $1.967_{\pm 0.514}$ | $0.478_{\pm 0.029}$ | $10.09_{\pm 1.798}$ | $0.551_{\pm 0.070}$ |
| **AGMM** | $\underline{5.941_{\pm 0.994}}$ | $\underline{0.274_{\pm 0.045}}$ | $1.556_{\pm 0.252}$ | $0.414_{\pm 0.033}$ | $13.84_{\pm 1.340}$ | $0.375_{\pm 0.018}$ |
| **DeepIV** | $19.13_{\pm 2.327}$ | $0.662_{\pm 0.021}$ | $2.065_{\pm 0.305}$ | $0.642_{\pm 0.024}$ | $40.79_{\pm 13.15}$ | $0.605_{\pm 0.031}$ |
| **DFIV** | $11.73_{\pm 0.894}$ | $0.563_{\pm 0.025}$ | $2.928_{\pm 0.500}$ | $0.476_{\pm 0.025}$ | $24.78_{\pm 3.108}$ | $1.247_{\pm 0.097}$ |
| **CBIV** | $11.61_{\pm 2.675}$ | $0.551_{\pm 0.116}$ | $6.540_{\pm 1.465}$ | $0.760_{\pm 0.195}$ | $11.37_{\pm 3.168}$ | $0.414_{\pm 0.058}$ |
| **CFRNet** | $6.600_{\pm 0.606}$ | $0.290_{\pm 0.041}$ | $3.155_{\pm 2.893}$ | $0.482_{\pm 0.166}$ | $9.305_{\pm 1.370}$ | $0.387_{\pm 0.079}$ |
| **DRCFR** | $6.410_{\pm 0.533}$ | $0.310_{\pm 0.027}$ | $0.866_{\pm 0.298}$ | $0.447_{\pm 0.034}$ | $9.329_{\pm 1.685}$ | $0.348_{\pm 0.025}$ |
| **VCNet** | $7.490_{\pm 0.289}$ | $0.309_{\pm 0.026}$ | $\underline{0.611_{\pm 0.128}}$ | $\underline{0.229_{\pm 0.031}}$ | $8.298_{\pm 2.338}$ | $\underline{0.263_{\pm 0.072}}$ |
| **CEVAE** | $9.899_{\pm 0.592}$ | $0.525_{\pm 0.053}$ | $4.585_{\pm 0.539}$ | $0.722_{\pm 0.038}$ | $21.88_{\pm 2.148}$ | $0.867_{\pm 0.061}$ |
| **TEDVAE** | $16.24_{\pm 0.379}$ | $0.702_{\pm 0.013}$ | $6.546_{\pm 0.768}$ | $0.691_{\pm 0.023}$ | $29.77_{\pm 2.955}$ | $0.764_{\pm 0.019}$ |
| **KerIRM** | $13.12_{\pm 2.589}$ | $0.479_{\pm 0.073}$ | $3.696_{\pm 0.978}$ | $0.649_{\pm 0.042}$ | $23.93_{\pm 2.944}$ | $0.627_{\pm 0.031}$ |
| **KerHRM** | $17.94_{\pm 3.808}$ | $0.547_{\pm 0.083}$ | $5.383_{\pm 1.588}$ | $0.581_{\pm 0.077}$ | $24.61_{\pm 2.974}$ | $0.659_{\pm 0.093}$ |
| **VANILLA** | $7.512_{\pm 1.048}$ | $0.348_{\pm 0.067}$ | $2.068_{\pm 1.917}$ | $0.510_{\pm 0.323}$ | $19.66_{\pm 14.98}$ | $0.656_{\pm 0.317}$ |
| **EnCounteR** | $\mathbf{4.816_{\pm 0.609}}$ | $\mathbf{0.210_{\pm 0.026}}$ | $\mathbf{0.582_{\pm 0.130}}$ | $\mathbf{0.188_{\pm 0.021}}$ | $\mathbf{5.751_{\pm 0.606}}$ | $\mathbf{0.186_{\pm 0.038}}$ |

**Results** In the linear simulation experiments (Figure 2), we employ three parametric estimators: the linear analytical solution (**LAS**) from Theorem 1, the **GMM** reformulation in Eq. 5, and our **EnCounteR** in Eq. 10. The LAS method relies on a substantial variation, $\beta_{T|X}^{(e_1)} - \beta_{T|X}^{(e_0)}$, and is limited to estimating $\phi_t$ using only a single $X$ variable. As data dimensions increase in Figure 2, the influence of variations in single $X$ on $T$ diminishes, which would introduce larger errors in $\varepsilon_{\text{CE}}$. Moreover, inaccuracies in estimating $\beta_{Y|X}^{(e_k)}$ and $\beta_{T|X}^{(e_k)}$ could exacerbate LAS errors by magnifying them further. To address the over-identification issue, we use GMM and EnCounteR reformulations to identify the causal parameter leveraging moments on residuals from full variables $X$. As shown in Figure 2, regardless of varying dimensions of $X$, both GMM and EnCounteR consistently exhibit robustness in estimating causal effects. Following Corollary 2, EnCounteR with novel moments (Eq. 8) yields more accurate estimates of causal parameters with lower variance. Furthermore, with varying encouragement proportions, $\rho = \{10\%, 20\%, 30\%, 50\%, 100\%\}$, our EnCounteR consistently performs well when $n_1 \geq n_0 \times 30\%$, greatly reducing the experiments costs and the computational expenses. Therefore, in subsequent studies, we set $n_k = n_0 \times 30\%$ for $k \geq 1$.

## 5.3 EXPERIMENTS ON COMPLEX DATASETS

**Datasets** In complex non-linear setting with heterogeneous treatment effects, we evaluate the EnCounteR method on Simulations (MULTs) and two real datasets (IHDP and ACIC2016). First, we synthesize **Simulations (MULTs) with Covariate Shifts** across various encouragements $e_k \in \{e_0, e_1, \cdots, e_K\}$: $X^{(e_k)} \sim \mathcal{N}(\mu_x^{(e_k)}, \sigma_x^{(e_k)})$ with fixed covariance 0.3, where $\mu_x^{(e_k)} \sim \text{Unif}(-0.2, 0.2)$ and $\sigma_x^{(e_k)} \sim \text{Unif}(0.7, 1.3)$, and $U_i^{(e_k)} \sim \mathcal{N}(0.3(X_{2i-1}^{(e_k)} + X_{2i}^{(e_k)}), 1)$, where subscript $i$ denotes the $i$-th variable in $U$. In the main experiments, guided by the findings in Section 5.2, we set observational data at $n_0 = 2,000$ and experimental data at $n_k = 600$ for $1 \leq k \leq K$, with parameters $K = 1$, $d_x = 5$, and $d_u = 3$. The treatment and outcome generation with non-linear multiplicative cross-terms for the MULT dataset is detailed in Appendix C. Moreover, we generate four MULT datasets with more encouragements ($K > 1$) and different sample sizes $n_k$, keeping $n_0 = 2,000$. Furthermore, to simulate complex scenarios, we add three non-linear terms into outcome functions to evaluate our EnCounteR, as outlined in Table 2.

We also apply two widely-adopted benchmarks: **IHDP** (Hill, 2011; Shalit et al., 2017) of 747 samples with 5 observed and 20 unmeasured confounders, **ACIC 2016** (Dorie et al., 2019) of $4,802$ samples with 12 observed and 46 unmeasured confounders. More details are provided in Appendix C.

**Main Results** We compare our method with baselines for estimating the counterfactual outcomes and CATE on the above datasets, each with 10 replications. The mean and standard deviation of $\varepsilon_{\text{CFR}}$ and $\varepsilon_{\text{PEHE}}$ are shown in Table 1, and the optimal and second-optimal performance are **bold** and

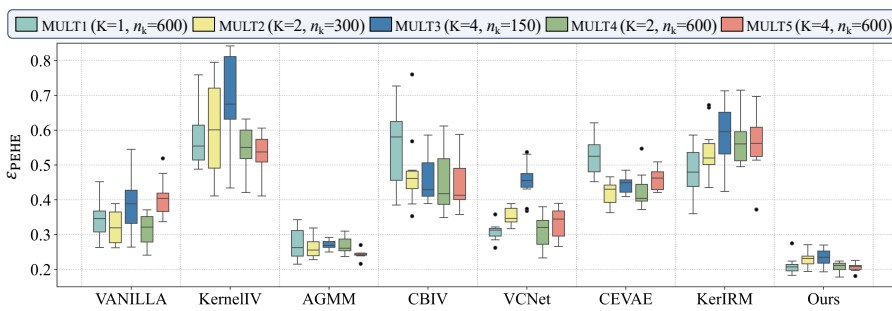

Figure 3: Box Plot of $\epsilon_{\text{PEHE}}$ in Simulation (MULTs): Varying Encouragements and Volume $K \times n_k$.

underlined, respectively. First, from Table 1, we can find that IV-based methods are limited in fully capturing exogenous variations in continuous treatments due to discrete encouragements, failing to precisely estimate causal effects as the exogeneity is insufficient for confounding effects. Second, covariate-based methods such as CFRNet, DRCFR, and VCNet also underperform because the unconfoundedness assumption is violated; as seen from the results, CEVAE and TEDVAE are prone to overfit; and methods including KerIRM and KerHRM fail to deal with unmeasured confounding and observed variables' entanglements. Third, the proposed EnCounteR outperforms all baselines across various datasets, achieving state-of-the-art performance. Compared to the second-optimal method, our EnCounteR on Simulation, IHDP, and ACIC datasets further reduce $\epsilon_{\text{CFE}}$ by 19%, 10%, and 31%, and $\epsilon_{\text{PEHE}}$ by 23%, 18%, and 29%, respectively. These results highlight the scalability of our method to complex data, demonstrating its potential for real applications.

**The scalability of our EnCounteR across varying encouragements $K$ and sub-data volume $n_k$** We evaluate EnCounteR's scalability with varying $K$ and data volumes ($K \times n_k = 600$, $n_0 = 2,000$) on MULT1, MULT2, and MULT3. As shown in Figure 3, larger $K$ and smaller $n_k$ lead to higher errors and variance. In contrast, on MULT1, MULT4, and MULT5 with fixed $n_k = 600$, increasing $K$ reduces variance without affecting mean error. These results suggest that EnCounteR's performance benefits from larger $n_k$ in at least one encouragement group and higher $K$, implying that employing a single encouragement with more samples can be more effective than using many small groups.

The supplementary experiments on additional non-linear scenarios, ablation study, and hyperparameter optimization are provided in Appendix D.1, Appendix D.2, and Appendix D.3, respectively.

## 6 CONCLUSION

Despite the growing body of literature on encouragement designs (EDs) for estimating causal effects, real-world applications often face challenges such as limited experimental data, non-random encouragement assignments, and a small number of encouragement strategies. To address these challenges, we introduce a generalized instrumental variables estimator called **En**couragement-based **Counter**factual **R**egression (**EnCounteR**), which provides identifiability guarantees and efficient methods for estimating CATE under positive encouragement experiments. EnCounteR enables accurate and low-variance estimation of treatment effects in both discrete and continuous treatment settings. Notably, it supports causal identification even when the number of instrumental variables is smaller than the number of treatments—a scenario traditionally considered problematic. By extending the boundaries of classical IV approaches, EnCounteR offers a robust framework for causal inference in complex and data-limited environments.

## REPRODUCIBILITY STATEMENT

Theoretical claims and required assumptions are clarified in Section 3.2, with proofs in Appendix A. More details of algorithm implementation and datasets are included in Appendices B-C.

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

## A THEOREMS AND PROOFS

### A.1 THE PROOFS OF THEOREMS 1

**Theorem 1.** *Under Assumptions 1 & 2, given datasets $\{\mathcal{D}^{(e_0)}, \mathcal{D}^{(e_1)}, \cdots, \mathcal{D}^{(e_K)}\}$ with different encouragements $\{e_0, e_1, \cdots, e_K\} \in \mathcal{E}$, the causal effect $\psi_t$ is identified by $\psi_t = \frac{\beta_{Y|X}^{(e_1)} - \beta_{Y|X}^{(e_0)}}{\beta_{T|X}^{(e_1)} - \beta_{T|X}^{(e_0)}}$, regardless of whether the treatment $T$ is binary, multi-valued, or continuous.*

*Proof.* Without loss of generality, we consider two datasets, $\mathcal{D}^{(e_0)}$ and $\mathcal{D}^{(e_1)}$, for the proof. First, we define $\beta_{Y|X}^{(e_k)}$, $\beta_{T|X}^{(e_k)}$, and $\beta_{U|X}^{(e_k)}$ for any $e_k$:

$$\beta_{Y|X}^{(e_k)} = \frac{\mathrm{Cov}\left(Y^{(e_k)}, X^{(e_k)}\right)}{\mathrm{Var}\left(X^{(e_k)}\right)}, \beta_{T|X}^{(e_k)} = \frac{\mathrm{Cov}\left(T^{(e_k)}, X^{(e_k)}\right)}{\mathrm{Var}\left(X^{(e_k)}\right)}, \beta_{U|X}^{(e_k)} = \frac{\mathrm{Cov}\left(U^{(e_k)}, X^{(e_k)}\right)}{\mathrm{Var}\left(X^{(e_k)}\right)} \quad (17)$$

Then, we can reformulate $\beta_{Y|X}^{(e_k)}$ as follows:

$$\beta_{Y|X}^{(e_k)} = \frac{\mathrm{Cov}\left(\psi_t T^{(e_k)} + \psi_x X^{(e_k)} + \psi_u U^{(e_k)}, X^{(e_k)}\right)}{\mathrm{Var}\left(X^{(e_k)}\right)} = \psi_t \beta_{T|X}^{(e_k)} + \psi_x + \psi_u \beta_{U|X}^{(e_k)}. \quad (18)$$

Given $\{X, U\} \perp\!\!\!\perp \mathcal{E}$, the covariance between $U^{(e_k)}$ and $X^{(e_k)}$, and hence $\beta_{U|X}^{(e_k)}$ remains constant across encouragements. Then, we define $b = \psi_x + \psi_u \beta_{U|X}^{(e_k)}$. For encouragements $e_0$ and $e_1$:

$$\beta_{Y|X}^{(e_0)} = \psi_t \beta_{T|X}^{(e_0)} + b, \quad \beta_{Y|X}^{(e_1)} = \psi_t \beta_{T|X}^{(e_1)} + b, \quad (19)$$

The causal effect is identified by $\psi_t = \frac{\beta_{Y|X}^{(e_1)} - \beta_{Y|X}^{(e_0)}}{\beta_{T|X}^{(e_1)} - \beta_{T|X}^{(e_0)}}$. $\qquad\qquad\square$

### A.2 THE PROOFS OF COROLLARY 2

**Corollary 2.** *Under Assumptions 3 & 4, the result of the estimated $\theta^*$ in Eq. 10 equals $h_\theta(T, X)$.*

*Proof.* Under Assumptions 3 and 4, Theorem 3 guarantees the existence of a unique solution $h_\theta(T, X)$, which accounts for the correlation between the additional noise and observed covariates $\mathbb{E}[\varepsilon(U) \mid X]$. Furthermore, moment condition 9 guarantees that the residual ($\epsilon = Y - \hat{Y}_\theta$) remains independent of encouragements ($e$). These conditions collectively enable us to minimize the loss function, $l(Y, \hat{Y}_\theta)$, to approximate $h_\theta(T, X)$ accurately. $\qquad\square$

## B IMPLEMENTATION DETAILS

In this paper, we use two-layer neural networks with ELU activation, with each layer containing $d_h$ hidden units, for both Counterfactual Regression $\hat{Y}_\theta = h_\theta(T, X)$ and $d_r$-dimensional Representation $R = r_\xi(X)$. We adopt full-batch training for the proposed EnCounteR algorithm, optimize it with the objective function 16, and set the maximum number of training epochs to 1,000. EnCounteR contains three hyperparameters, i.e., $d_r \in \{1, 5, 8, 10, 12, 20\}$, $d_h \in \{16, 32, 64, 128, 256\}$, and $\alpha \in \{1, 2, 5, 12, 15, 20\}$. We utilize the minimum regression error on the validation dataset to optimize hyper-parameters. The pseudocode is placed in Algorithm 1.

---

**Algorithm 1** EnCounteR: Encouragement-Based Counterfactual Regression

---

**Input:** Encouragement designs $\mathcal{D} = \{\mathcal{D}^{(e_k)}\}_{e_k \in \{e_0, e_1, \cdots, e_K\}}$, each with $\mathcal{D}^{(e_k)} = \{x_i^{(e_k)}, t_i^{(e_k)}, y_i^{(e_k)}\}_{i=1}^{n_k}$; Hyper-parameters $\{d_h, d_r, \alpha\}$; Trainable Weighting Vectors $\omega^{(e_k)} = \frac{1+3\sigma(w^{(e_k)})}{\sum_i^{n_k}(1+3\sigma(w^{(e_k)}))}$ with default $w^{(e_k)} = 1$; Counterfactual Regression Network $h_\theta(\cdot)$ with Trainable Parameters $\theta$; Adversarial Representation Network $r_\xi(\cdot)$ with Trainable Parameters $\xi$; Reweighting-Training-Epoch $\mathcal{I}_1 = 10$; Adversarial-Training-Epoch $\mathcal{I}_2 = 100$; Full-Training-Epoch $\mathcal{I}_3 = 1,000$.

**Output:** Counterfactual Outcome Function $\hat{Y}_\theta(t, X) = h_\theta(do(t), X)$, and Conditional Average Treatment Effect $\text{CATE}(t, X) = \hat{Y}_\theta(t, X) - \hat{Y}_\theta(0, X)$.

**Loss function:** $\mathcal{L}_\omega$ in Eq. 11, $\mathcal{L}_R$ in Eq. 14, and $\mathcal{L} = \mathcal{L}_{\text{REG}} + \alpha(\mathcal{L}_\mathcal{E} + \mathcal{L}_X + \mathcal{L}_R)$ in Eq. 16.

**Reweighting for Covariate Balance:**

**for** itr $= 1$ **to** $\mathcal{I}_1$ **do**

    $\mathbb{E}_\omega[X^{(e_k)}] = \sum_i^{n_k} \omega_i^{(e_k)} x_i^{(e_k)},$

    $\text{Cov}_\omega[X^{(e_k)}] = \left[\sum_i^{n_k} \omega_i^{(e_k)}(x_{i,a}^{(e_k)} - \mathbb{E}_\omega[X_a^{(e_k)}])(x_{i,b}^{(e_k)} - \mathbb{E}_\omega[X_b^{(e_k)}])\right]_{1 \le a, b \le d_x}$, where $a$ denotes $a$-th variable in $X$,

    $\mathcal{L}_\omega = \sum_{j \ne k}(\mathbb{E}_\omega[X^{(e_j)}] - \mathbb{E}_\omega[X^{(e_k)}])^2 + (\text{Cov}_\omega[X^{(e_j)}] - \text{Cov}_\omega[X^{(e_k)}])^2,$

    update $\omega \leftarrow \text{Adam}\{\mathcal{L}_\omega\}$ using Adaptive Moment Estimation.

**end for**

**Counterfactual Regression:**

**for** itr $= 1$ **to** $\mathcal{I}_3$ **do**

    **if** itr $\le \mathcal{I}_2$ **then**

        $\{X^{(e_k)}, T^{(e_k)}, Y^{(e_k)}\}_{0 \le k \le K} \rightarrow \mathcal{L}_R = g_R'(\theta, \xi) \cdot W_R \cdot g_R(\theta, \xi),$

        update $\xi \leftarrow \text{Adam}\{-\widetilde{\mathcal{L}_R}\}$ in representation network $h_\theta(\cdot)$ using Adaptive Moment Estimation.

    **end if**

    $\{X^{(e_k)}, T^{(e_k)}, Y^{(e_k)}\}_{0 \le k \le K} \rightarrow \mathcal{L} = \mathcal{L}_{\text{REG}} + \alpha(\mathcal{L}_\mathcal{E} + \mathcal{L}_X + \mathcal{L}_R),$

    update $\theta \leftarrow \text{Adam}\{\mathcal{L}\}$ in counterfactual regression $h_\theta(\cdot)$ using Adaptive Moment Estimation

**end for**

---

**Hardware used**: Ubuntu 16.04.3 LTS operating system with 2 * Intel Xeon E5-2660 v3 @ 2.60GHz CPU (40 CPU cores, 10 cores per physical CPU, 2 threads per core), 256 GB of RAM, and 4 * GeForce GTX TITAN X GPU with 12GB of VRAM.

**Software used**: Python 3.7.15 with TensorFlow 1.15.0, Pytorch 1.7.1, and NumPy 1.18.0.

## C  DESCRIPTION OF USED COMPLEX DATASETS

In this section, we present an overview of the complex datasets used in our study. Within the main text, we evaluate the performance of the EnCounteR method on five simulations (MULTs) and two real-world datasets (IHDP and ACIC2016), with results shown in Tables 1 & 3 and Figure 4. To further evaluate the robustness of the EnCounteR algorithm in more complex scenarios, we introduce three additional non-linear terms into the outcome function, with these extended experiments presented in Table 2. Next, we provide a detailed description of these datasets.

**Simulations**  Firstly, we introduce the generation process of **Simulations (MULTs) with Covariate Shifts** across different encouragements $e_k \in \{e_0, e_1, \cdots, e_K\}$. For each encouragements $e_k$, we generate the observed covariates by $X^{(e_k)} \sim \mathcal{N}(\mu_x^{(e_k)}, \Sigma_x^{(e_k)}), \mu_x^{(e_k)} \sim \text{Unif}(-0.2, 0.2)$ with:

$$\Sigma_x = \begin{pmatrix} \sigma_x^{(e_k)} & 0.3 & \cdots & 0.3 \\ 0.3 & \sigma_x^{(e_k)} & \cdots & 0.3 \\ \vdots & \vdots & \ddots & \vdots \\ 0.3 & 0.3 & \cdots & \sigma_x^{(e_k)} \end{pmatrix},$$

$$\sigma_x^{(e_k)} \sim \text{Unif}(0.7, 1.3).$$

where $d_x$ denotes the dimension of $X$, and $X_{1\cdots d_x} = \{X_1, X_2, \cdots, X_{d_x}\}$. Then, the unmeasured confounders would be $U_i^{(e_k)} \sim \mathcal{N}(0.3(X_{2i-1}^{(e_k)} + X_{2i}^{(e_k)}), 1)$, where subscript $i$ denotes the $i$-th variable in $U$ and $i \in \{1, 2, \cdots, d_u\}$. In the main experiments, guided by the findings in Section 5.2, we set observational data at $n_0 = 2{,}000$ and experimental data at $n_k = 600$ for $1 \leq k \leq K$, with parameters $K = 1$, $d_x = 5$, and $d_u = 3$. Define $C \in \mathbb{R}^{d_x + d_u}$ as the concatenation of all confounders $X$ and $U$, we generate the treatments and outcomes with non-linear multiplicative cross-terms as follows:

$$
T^{(e_k)} = \left| \sum_{i=1}^{d_x+d_u-1} \left[ \phi_i^{(e_k)} C_i^{(e_k)} C_{i+1}^{(e_k)} \right] + \sum_{i=1}^{d_x+d_u} \left[ \phi_i^{(e_k)} C_i^{(e_k)} \right] \right|,
$$

$$
Y_{\text{MULT}}^{(e_k)} = T^{(e_k)} \times (0.5 + X_0^{(e_k)}) + \sum_{i=1}^{d_x-1} \left[ \psi_i X_i^{(e_k)} X_{i+1}^{(e_k)} \right]
$$

$$
+ \sum_{i=1}^{d_u-1} \left[ \psi_i U_i^{(e_k)} U_{i+1}^{(e_k)} \right] + \sum_{i=1}^{d_x+d_u} \left[ \psi_i C_i^{(e_k)} \right],
$$

where, the coefficients $\{\phi_{1\cdots(d_x+d_u)}^{(e_k)}, \psi_{1\cdots(d_x+d_u)}\}$ are independently sampled from a Uniform distribution $\text{Unif}(0,1)$ In the above equations, we name this simulation with non-linear multiplicative cross-terms as MULT.

**MULTs with Varying Encouragements $K$ and Data Volume $K \times n_k$ for $1 \leq k \leq K$**  We name the above original dataset as MULT1 with $K = 1$, $n_k = 600$ and total volume $K \times n_k = 600$ for $k > 0$. To further explore the effects of increased encouragements and varying data volumes, we generate four additional datasets with more encouragements ($K > 1$), keeping the observational data size at $n_0 = 2{,}000$ and varying the sample sizes $n_k$ for $1 \leq k \leq K$. Keeping a fixed total volume of encouragement data at $K \times n_k = 600$ for $k > 0$, we construct two additional datasets: MULT2 with $K = 2$ and $n_k = 300$ for $k > 0$, and MULT3 with $K = 4$ and $n_k = 150$ for $k > 0$. This allows us to analyze the effects of varying the number of encouragements while keeping the total volume of encouragement data constant. With fixed size $n_k = 600$, we create two additional datasets: MULT4 with $K = 2$ and a total encouragement data volume of $K \times n_k = 1{,}200$ for $k > 0$, and MULT5 with $K = 4$ and a total encouragement data volume of $K \times n_k = 2{,}400$ for $k > 0$. This enables us to conduct a comprehensive analysis of the influence of varying numbers of encouragements and total encouragement data volumes on our study's outcomes.

**Simulations with Additional Non-linear Terms: POLY, ABS and SIN**  To simulate real-world conditions, we add three additional non-linear terms in the outcome function for assessing the EnCounteR algorithm:

$$
Y_{\text{POLY}}^{(e_k)} = Y_{\text{MULT}}^{(e_k)} + T^{(e_k)} \times \left( X_1^{(e_k)} \right)^2 + \sum_{i=1}^{d_x+d_u} \left[ \psi_i \left( C_i^{(e_k)} \right)^2 \right] / d_x,
$$

$$
Y_{\text{ABS}}^{(e_k)} = Y_{\text{MULT}}^{(e_k)} + T^{(e_k)} \times \left| X_1^{(e_k)} \right| + \sum_{i=1}^{d_x+d_u} \left[ \psi_i \left| C_i^{(e_k)} \right| \right] / d_x,
$$

$$
Y_{\text{SIN}}^{(e_k)} = Y_{\text{MULT}}^{(e_k)} + T^{(e_k)} \times \sin\left( X_1^{(e_k)} \right) + \sum_{i=1}^{d_x+d_u} \left[ \psi_i \sin\left( C_i^{(e_k)} \right) \right] / d_x.
$$

We name these three simulations with additional non-linear terms as POLY, ABS and SIN, respectively. For each data, we combine $\{\mathcal{D}^{e_k}\}_{e_k \in \mathcal{E}}$ as training data, reserving 10%-30% of $\mathcal{D}^{e_0}$ as validation data, and generate 20,000 additional samples with random treatments $do(t) \sim \text{Unif}[0,1]$ and the outcomes $Y(do(t), X)$ as testing data.

**Real-World Data**  Although massive open online courses (MOOCs) like Coursera, edX, and Udacity bring a deluge of data about student behavior in classrooms (Breslow et al., 2013; Kizilcec et al., 2014; Reich, 2015), due to concerns over information privacy, we lack access to complete data on student behavior in MOOCs. Furthermore, based on the publicly available data, specifically `https://doi.org/10.7910/DVN/26147` and `http://moocdata.cn/data/user-activity`,

Table 2: Results (mean$_{\pm\text{std}}$) on Complex Simulation with Additional POLY, ABS and SIN Terms.

| | POLY | | ABS | | SIN | |
|---|---|---|---|---|---|---|
| Methods | $\epsilon_{\textbf{CFE}}$ | $\epsilon_{\textbf{PEHE}}$ | $\epsilon_{\textbf{CFE}}$ | $\epsilon_{\textbf{PEHE}}$ | $\epsilon_{\textbf{CFE}}$ | $\epsilon_{\textbf{PEHE}}$ |
| **KernelIV** | $25.31_{\pm1.906}$ | $0.585_{\pm0.040}$ | $21.44_{\pm2.115}$ | $0.593_{\pm0.106}$ | $19.67_{\pm2.625}$ | $0.644_{\pm0.189}$ |
| **AGMM** | $6.502_{\pm0.954}$ | $0.294_{\pm0.022}$ | $6.133_{\pm0.896}$ | $0.256_{\pm0.024}$ | $6.674_{\pm1.018}$ | $0.281_{\pm0.030}$ |
| **CBIV** | $11.64_{\pm4.388}$ | $0.506_{\pm0.102}$ | $10.01_{\pm2.869}$ | $0.499_{\pm0.123}$ | $10.03_{\pm2.921}$ | $0.538_{\pm0.142}$ |
| **VCNet** | $7.628_{\pm0.605}$ | $0.301_{\pm0.035}$ | $7.083_{\pm0.532}$ | $0.283_{\pm0.044}$ | $8.384_{\pm0.317}$ | $0.341_{\pm0.023}$ |
| **CEVAE** | $11.93_{\pm1.212}$ | $0.559_{\pm0.032}$ | $10.69_{\pm0.770}$ | $0.526_{\pm0.050}$ | $10.74_{\pm0.417}$ | $0.556_{\pm0.035}$ |
| **KerIRM** | $19.52_{\pm2.365}$ | $0.506_{\pm0.057}$ | $16.35_{\pm3.090}$ | $0.488_{\pm0.061}$ | $15.24_{\pm2.957}$ | $0.495_{\pm0.080}$ |
| **VANILLA** | $8.362_{\pm0.972}$ | $0.352_{\pm0.080}$ | $8.425_{\pm0.929}$ | $0.343_{\pm0.081}$ | $7.708_{\pm1.171}$ | $0.348_{\pm0.036}$ |
| **EnCounteR** | $\mathbf{5.294_{\pm0.434}}$ | $\mathbf{0.214_{\pm0.033}}$ | $\mathbf{5.029_{\pm0.446}}$ | $\mathbf{0.226_{\pm0.026}}$ | $\mathbf{4.840_{\pm0.616}}$ | $\mathbf{0.222_{\pm0.033}}$ |

we cannot construct complete encouragement data for evaluating our algorithm. Therefore, similar to previous work (Shalit et al., 2017; Yao et al., 2021), we validate our algorithm on the IHDP and ACIC2016 datasets.

**IHDP** The Infant Health and Development Program (IHDP[1]) (Hill, 2011) studies the effect of specialist home visits on the future cognitive test scores of premature infants, which comprises 747 units, with 139 in the treated group and 608 in the control group. There are 25 pre-treatment variables ($C \in \mathbb{R}^{25}$) related to the children and their mothers. In the IHDP study, to create multi-encouragement data, the large control group is used as $\mathcal{D}^{(e_0)}$, and the small treated group as $\mathcal{D}^{(e_1)}$. We select $d_x = 5$ continuous variables from the IHDP as observed covariates and use the expected potential outcomes $m_0$ for control outcomes and $m_1$ for treated outcomes as unmeasured confounding effects from the remaining $d_u = 20$ unmeasured variables. The encouraged treatments are from $T^{(e_k)} = |\sum_{i=1}^{d_x-1}[\phi_i^{(e_k)} X_i^{(e_k)} X_{i+1}^{(e_k)}] + \sum_{i=1}^{d_x}[\phi_i^{(e_k)} X_i^{(e_k)}] + m_0^{(e_k)}|$, and outcomes are determined by $Y_{\text{IHDP}}^{(e_k)} = T^{(e_k)} \times (0.5 + X_0^{(e_k)}) + \sum_{i=1}^{d_x-1}[\psi_i X_i^{(e_k)} X_{i+1}^{(e_k)}] + \sum_{i=1}^{d_x}[\psi_i X_i^{(e_k)}] + m_1^{(e_k)}$. From the control group $\mathcal{D}^{(e_0)}$, We split 75 samples for validation data and another 75 for pre-testing data, leaving $n_0 = 458$ samples as encouragement data with $e_0$, while maintaining $n_1 = 139$ in the treated group. The pre-testing data is replicated 100 times to create 7,500 samples with random treatments $do(t) \sim \text{Unif}[0, 1]$, and for these samples, we regenerate the corresponding outcomes $Y(do(t), X)$ to be used as testing.

**ACIC2016** The 2016 Atlantic Causal Inference Challenge (ACIC 2016[2]) (Dorie et al., 2019) holds the causal inference data analysis challenge, which creates 4,802 units, with 858 in the treated group and 3,944 in the control group. The two expected potential outcomes are $m_0$ for control outcomes and $m_1$ for treated outcomes. The covariates are real-world data from the full Infant Health and Development Program dataset, which consists of 58 variables ($C \in \mathbb{R}^{58}$). In the above ACIC study, we use the large control group as $\mathcal{D}^{(e_0)}$, and the small treated group as $\mathcal{D}^{(e_1)}$. We select $d_x = 12$ continuous variables from the ACIC as observed covariates and use the expected potential outcomes $m_0$ and $m_1$ as unmeasured confounding effects from the remaining $d_u = 46$ unmeasured variables. The encouraged treatments are from $T^{(e_k)} = |\sum_{i=1}^{d_x-1}[\phi_i^{(e_k)} X_i^{(e_k)} X_{i+1}^{(e_k)}] + \sum_{i=1}^{d_x}[\phi_i^{(e_k)} X_i^{(e_k)}] + 4m_0^{(e_k)}|$, and outcomes are determined by $Y_{\text{IHDP}}^{(e_k)} = T^{(e_k)} \times (0.5 + X_0^{(e_k)}) + \sum_{i=1}^{d_x-1}[\psi_i X_i^{(e_k)} X_{i+1}^{(e_k)}] + \sum_{i=1}^{d_x}[\psi_i X_i^{(e_k)}] + 4m_1^{(e_k)}$. From the control group $\mathcal{D}^{(e_0)}$, We split 480 samples for validation data and 480 for pre-testing data, leaving $n_0 = 2,984$ samples as encouragement data, while maintaining $n_1 = 858$ in the treated group. We then replicate the pre-testing data 20 times, creating 9,600 samples with random treatments $do(t) \sim \text{Unif}[0, 1]$, and regenerate outcomes $Y(do(t), X)$ for testing.

---

[1] IHDP datasets are available at: https://www.fredjo.com/.

[2] ACIC 2016 datasets are available at: https://github.com/vdorie/aciccomp/tree/master/2016.

Table 3: Ablation Study of EnCounteR Framework on Simulation, IHDP, and ACIC Datasets. EnCounteR is composed by four core modules: (a) $\omega$: Sample Reweighting Module in Eq. 11; (b) $\mathcal{L}_\mathcal{E}$: Encouragement-Independent Moments in Eq. 12; (c) $\mathcal{L}_X$: Covariate-Independent Moments in Eq. 13; (d) $\mathcal{L}_R$: Adversarial Representation-Independent Moments in Eq. 14.

| EnCounteR | Modules | | | | Simulation (MULT) | | IHDP | | ACIC | |
|---|---|---|---|---|---|---|---|---|---|---|
| | $+\omega$ | $+\mathcal{L}_\mathcal{E}$ | $+\mathcal{L}_X$ | $+\mathcal{L}_R$ | $\epsilon_{\text{CFE}}$ | $\epsilon_{\text{PEHE}}$ | $\epsilon_{\text{CFE}}$ | $\epsilon_{\text{PEHE}}$ | $\epsilon_{\text{CFE}}$ | $\epsilon_{\text{PEHE}}$ |
| $\mathcal{L}_{\text{REG}}$ | | | | | $7.512_{\pm 1.048}$ | $0.348_{\pm 0.067}$ | $2.068_{\pm 1.917}$ | $0.510_{\pm 0.323}$ | $19.66_{\pm 14.98}$ | $0.656_{\pm 0.317}$ |
| $\mathcal{L}_{\text{REG}}$ | ✓ | | | | $7.311_{\pm 1.020}$ | $0.344_{\pm 0.056}$ | $1.565_{\pm 0.984}$ | $0.452_{\pm 0.208}$ | $14.78_{\pm 6.163}$ | $0.587_{\pm 0.197}$ |
| $\mathcal{L}_{\text{REG}}$ | ✓ | ✓ | | | $7.841_{\pm 1.487}$ | $0.343_{\pm 0.082}$ | $1.306_{\pm 0.521}$ | $0.378_{\pm 0.124}$ | $12.31_{\pm 5.429}$ | $0.457_{\pm 0.126}$ |
| $\mathcal{L}_{\text{REG}}$ | ✓ | ✓ | ✓ | | $5.191_{\pm 0.533}$ | $0.224_{\pm 0.036}$ | $0.665_{\pm 0.213}$ | $0.215_{\pm 0.066}$ | $12.58_{\pm 5.421}$ | $0.476_{\pm 0.121}$ |
| $\mathcal{L}_{\text{REG}}$ | ✓ | ✓ | | ✓ | $5.733_{\pm 0.816}$ | $0.259_{\pm 0.036}$ | $0.710_{\pm 0.156}$ | $0.229_{\pm 0.024}$ | $5.689_{\pm 0.669}$ | $0.204_{\pm 0.022}$ |
| $\mathcal{L}_{\text{REG}}$ | | ✓ | ✓ | ✓ | $4.847_{\pm 0.607}$ | $0.220_{\pm 0.037}$ | $0.641_{\pm 0.203}$ | $0.199_{\pm 0.025}$ | $6.067_{\pm 0.927}$ | $0.218_{\pm 0.036}$ |
| $\mathcal{L}_{\text{REG}}$ | ✓ | ✓ | ✓ | ✓ | $\mathbf{4.816_{\pm 0.609}}$ | $\mathbf{0.210_{\pm 0.026}}$ | $\mathbf{0.582_{\pm 0.130}}$ | $\mathbf{0.188_{\pm 0.021}}$ | $\mathbf{5.751_{\pm 0.606}}$ | $\mathbf{0.186_{\pm 0.038}}$ |

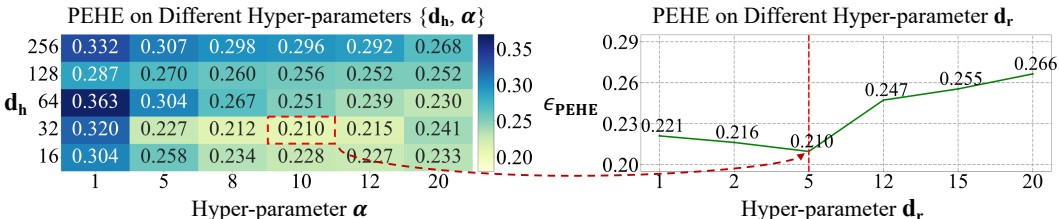

Figure 4: Hyper-Parameter Optimization: The minimum regression error on the validation data implies the optimal hyper-parameters. The optimal hyper-parameters are $d_h = 32, d_r = 5, \alpha = 10$.

# D  Supplementary Experiments

## D.1  The scalability of our EnCounteR across varying non-linear

We evaluate EnCounteR on outcome functions incorporating POLY, ABS, and SIN non-linear terms, compare it against advanced IV-based methods (KernelIV, AGMM, CBIV) and covariate-based methods (VCNet, CEVAE, KerIRM). As shown in Table 2, while most traditional methods underperform, including some worse than direct regression (VANILLA), only AGMM and VCNet show improvements. Compared to the optimal-second AGMM algorithm, our EnCounteR further reduces the $\epsilon_{\text{CFE}}$ by 18%, 18%, and 27% and the $\epsilon_{\text{PEHE}}$ by 27%, 11%, and 21% on the Simulation, IHDP, and ACIC datasets, respectively. Our EnCounteR exhibits robust and outstanding performance.

## D.2  Ablation Studies

EnCounteR is composed by four core modules: (a) $\omega$: Sample Reweighting Module in Eq. 11; (b) $\mathcal{L}_\mathcal{E}$: Encouragement-Independent Moments in Eq. 12; (c) $\mathcal{L}_X$: Covariate-Independent Moments in Eq. 13; (d) $\mathcal{L}_R$: Adversarial Representation-Independent Moments in Eq. 14. Table 3 reports the effects of each module of the EnCounteR by conducting ablation experiments on Simulation, IHDP and ACIC datasets. From Tables 1 and Table 3, we can draw the following conclusions: (I) Each component in our EnCounteR is essential, since missing any one of them would confuse the encouragement learning and damage the performance of potential outcome prediction and conditional average treatment estimation on three datasets. (II) When all components are fully utilized in EnCounteR, our method achieves optimal performance in causal effect estimation. The results demonstrate that each component of EnCounteR is crucial for estimating causa effects.

## D.3  The Optimization of Hyper-Parameters

In this paper, we adopt the minimum counterfactual regression error $\varepsilon_{\text{CFR}}$ on the validation data to determine the optimal hyper-parameters $\{d_h, d_r, \alpha\}$. Our approach follows this strategy: firstly, we search for $d_h \in \{16, 32, 64, 128, 256\}$ and $\alpha \in \{1, 2, 5, 12, 15, 20\}$, while fixing $d_r = d_x$, corresponding to the minimum validation error $\varepsilon_{\text{CFR}}$. Then, fixing the optimal $d_h$ and $\alpha$, we search for $d_r \in \{1, 5, 8, 10, 12, 20\}$ corresponding to the minimum validation error $\varepsilon_{\text{CFR}}$ again. Taking the

Table 4: Optimal Parameters on MULT, POLY, ABS, SIN, IHDP and ACIC Datasets.

| Params | MULT | POLY | ABS | SIN | IHDP | ACIC |
|---|---|---|---|---|---|---|
| $\alpha$ | 10 | 8 | 10 | 8 | 5 | 8 |
| $d_h$ | 32 | 128 | 128 | 32 | 128 | 32 |
| $d_r$ | 5 | 2 | 5 | 12 | 15 | 12 |
| $\epsilon_{PEHE}$ | $0.210_{\pm 0.026}$ | $0.214_{\pm 0.033}$ | $0.226_{\pm 0.026}$ | $0.222_{\pm 0.033}$ | $0.188_{\pm 0.021}$ | $0.186_{\pm 0.038}$ |

main experiment MULT as an example, as depicted in Figure 4, we determine the hyper-parameters that correspond to the smallest $\varepsilon_{CFR}$ on the validation, which also indicates the smallest $\varepsilon_{PEHE}$ on MULT. The optimal hyper-parameters are $d_h = 32, d_r = 5, \alpha = 10$ for MULT. Table 4 shows the optimal hyper-parameters for each dataset.

## E  LARGE LANGUAGE MODEL USAGE STATEMENT

We used Large Language Models (LLMs) to polish up our writing. The models provided suggestions for grammatical corrections, sentence adjustments, and more precise vocabulary choices. We carefully reviewed and integrated these suggestions to improve the readability of the manuscript.

