# OpenReview forum: "Generalized Encouragement-Based Instrumental Variables for Counterfactual Regression"
_ICLR.cc/2026/Conference — ICLR 2026 Conference Withdrawn Submission_

### Official Review · Reviewer_tLCN · 2025-10-25

**Soundness:** 1
**Presentation:** 2
**Contribution:** 2
**Rating:** 2
**Confidence:** 4

**Summary:**

The paper presents an interesting approach that utilizes encouragement designs (EDs) to identify Conditional Average Treatment Effects (CATE), particularly in situations where randomized controlled trials (RCTs) are impractical or difficult to implement. The method is evaluated on both synthetic and real-world datasets.

**Strengths:**

The idea of leveraging encouragement designs (EDs) as instrumental variables (IVs) to identify CATE is novel and offers a practical solution in real-world settings.

**Weaknesses:**

Mathematical Rigor:
The mathematical framework is not sufficiently rigorous in some places. For example, in Assumption 2, the epsilon is defined as a set, and it is not explained how this set would be independent of the data X and U. Additionally, Theorem 1's denominator cannot be zero, but this condition is not properly addressed or discussed in the paper.

Several claims made in the paper lack precision:
Paragraph 2 and 3 “violating the condition that the support set of IVs must exceed that of treatments” is not accurate. A binary IV can indeed work with a continuous treatment as long as the relevance condition holds. In fact, as long as the instrument is valid, binary IVs are commonly used in situations with continuous treatments, and they don’t necessarily pose a problem as long as there’s sufficient correlation between the IV and the treatment. The idea of identifying the Local Average Treatment Effect (LATE) primarily applies in cases where the instrument is binary, but as you pointed out, it's also a non-parametric concept and applies more generally. If the model is linear, two-stage least squares (2SLS) with a binary IV will indeed recover the correct treatment effect, assuming the assumptions (relevance, exclusion restriction, and exogeneity) are satisfied.

Page 4, Theorem Contribution:
The paper does not clarify the number of IVs in their setting. For example, in the example in Figure 1, how many IVs are there—one or two? You can certainly generate dummy variables for the two treatment arms, Verbal Praise and Points Reward. However, since there is only one treatment (Forum Time) in the example, the number of IVs should exceed the number of treatments. This issue appears in several places throughout the paper.

Key Identification assumption hard to satisfy. In Theorem 1, the key assumption is cov(T, X) should be different under different encouragement design, but cov(U, X) should be the same. If we assume T = a + b*x + u, U = c + d*x + e, then this assumption requires that b^(e0) != b^(e1) and d^(e0) = d^(e1). For example, in Figure 1, T is forum time, X and U could be student characteristics such as IQ and time spent on homework. What kind of encouragement would change the b (the effect of IQ on forum time) but not d (the effect of IQ on homework)?

**Questions:**

Why did you choose to formulate the encouragement design as {e0, e1} instead of using a binary indicator E=1 or 0? It would be helpful to explain why this formulation is preferred and how it affects the model's assumptions or the interpretation of the results.

Could you provide convincing examples where the assumptions in Theorem 1 hold in practice?

---

### Official Review · Reviewer_1zZp · 2025-10-30

**Soundness:** 2
**Presentation:** 3
**Contribution:** 2
**Rating:** 4
**Confidence:** 3

**Summary:**

This paper proposes EnCounteR, a method for estimating causal effects in encouragement designs where treatments can be continuous but encouragements are limited and discrete. The method addresses three practical challenges: non-random encouragement assignments, limited experimental data, and cases where the number of available encouragements is smaller than the dimension of treatments. The theoretical contribution shows that causal effects can be identified even when traditional IV conditions (requiring more instruments than treatments) are violated, by using heterogeneity in observed covariates across encouragement groups. The method combines observational data with encouragement experiments (somewhat similar to semi-supervised learning), uses sample reweighting to handle covariate shifts across encouragement groups, and employs three types of moment constraints within an adversarial learning framework.

**Strengths:**

* The paper is clearly written with good motivation from applications in online education platforms.

* The paper tackles real-world issues in encouragement designs including non-random assignment, limited sample sizes, and the case where the number of discrete encouragement policies is much smaller than the dimension of continuous treatments.

* The authors provide formal identification results, progressing from Theorem 1 for the linear case to non-linear settings in Theorem 3 and Corollary 1.

* The experiments include multiple synthetic datasets (MULT, POLY, ABS, SIN) with varying complexity levels and benchmarks on IHDP and ACIC datasets, demonstrating consistent improvements over existing baselines.

* The paper provides a detailed algorithm (Algorithm 1) with clear implementation details, facilitating reproducibility.

**Weaknesses:**

* Theorem 3 appears to be a result from the literature (Newey & Powell, 2003) rather than a contribution of this paper. While this is mentioned in a proof comment, it should be clearly stated when the theorem is first presented to avoid confusion about the paper's contributions.

* It's unclear to me what the real technical contribution is here. It seems like the treatment effect is identifiable from existing results unless the encouragements "only introduce a minor exogenous disturbance to the continuous treatment that is too small to accurately estimate CATE" (L239). Hence, the real technical novelty is a slight reformulation of the moment conditions in Equations (11)-(13).

* The assumption in the covariate balancing of the samples between different encouragements to "exhibit slight covariate shifts" (L279) seems to somewhat contradict the main premise of the paper of dealing with encouragements acting on different subgroups.

**Minor:**

* Theorem 1 should only have two encouragement regimes in the datasets.
* L163: typo

**Questions:**

1. What is the precise technical barrier that EnCounteR overcomes? Could you provide a concrete example where existing IV methods (e.g., using Theorem 3 directly) would fail to identify the treatment effect, but EnCounteR succeeds? This would help clarify the core contribution beyond the reformulation of moment conditions.

2. How do you reconcile the contradiction between non-random encouragement assignment and the "slight covariate shift" assumption? If encouragements are assigned to pre-existing groups with potentially substantial differences (e.g., classes with different teacher quality), how can we assume only slight covariate shifts? What happens to the identification results when this assumption is violated?

3. What is the formal justification for the covariate balancing approach? The paper uses sample reweighting to match means and covariances across encouragement groups (Eq. 14), but doesn't provide theoretical guarantees that this addresses the non-random assignment problem. Could you provide either theoretical justification or empirical evidence that this approach successfully handles selection bias?

4. Why not evaluate on real encouragement design data? The motivation discusses online education platforms extensively, but the experiments only use semi-synthetic benchmarks (IHDP, ACIC). Have you attempted to apply EnCounteR to actual encouragement design studies? If not, what are the practical barriers?

5. How does the method perform with very weak encouragements? Given that the motivation for combining Theorems 1 and 3 is to handle cases where discrete encouragements provide minimal treatment variation, could you provide specific experiments or analysis showing performance degradation as encouragement strength decreases, and how EnCounteR maintains identification in these challenging cases?

---

### Official Review · Reviewer_QW5W · 2025-11-06

**Soundness:** 3
**Presentation:** 2
**Contribution:** 3
**Rating:** 6
**Confidence:** 2

**Summary:**

The authors focus on estimating treatment effect in encouragement designs.  Encouragement designs are common in situations where RCTs can't be performed.  However, especially in situations with continuous treatment, using encouragement as an instrument for effect estimation is challenging since there are far fewer possible values for encouragement than treatment.  First, the authors propose an identification method for estimating CATE using discrete encouragement information.  They then leverage this to propose an IV estimator for estimating causal effects even when the domain of encouragement is smaller than that of treatment.  They evaluate the performance of their method on both synthetic and semi-synthetic data and show that it consistently out-performs other estimation methods.

**Strengths:**

The paper is very well-written.  The prose is clear and the notation is precise and consistent.  The problem being tackled is interesting and relevant.  The methodology employed appears sound, and the experimental results compare a nice range of methods.

**Weaknesses:**

While broadly familiar with instrumental variables, I'm unfortunately not overly well-versed in the technical details of IV estimation.  While I was able to follow the majority of the paper, I found page 6 (Sections 4.1 - 4.3) fairly hard to follow.  Section 4 seems to jump right into the technical details without providing much high level intuition and summary.  It's unclear to me, though, how much of this is is a result of my lack of technical knowledge in this area and how much is actually a result of poor explanation on the part of the authors.  I'm hoping the reviews of the other reviewers will help me assess if Section 4 is sufficiently descriptive.  For now, I'm keeping my review score fairly non-committal and will aim to engage with the authors' response and the comments of the other reviewers to improve my understanding and allow for a more accurate score.

In Section 2, it's clear why "encouragement designs" and "instrumental variables" fall under related work.  "Multiple environments", however, feels far more tangential.  Is "multiple environments" referring to the encouragement vs observational datasets?  If so, the way the observational data is described in this paper makes it seem more like a different setting rather than a different environment.

On line 114, you define $n_k$ as the sample size of encouragement $e_k$.  However, later on line 138, you instead use $n^{(e_k)}$.  Is this the same thing?  If so, they notation should match, and if not, clarify what it means here.

In Section 3.2.3, you provide Equation (6) and describe it as "the transformed outcome."  However, from what I can tell, this is the first time you're using the term $h_\theta$, and there's no further description of what "the transformed outcome" means.

This is minor, but on line 362, you define the Precision in Estimation of Heterogeneous Effect.  However, since you're taking the square root, which isn't part of the standard definition of PEHE, you should probably call it the "root Precision in Estimation of Heterogeneous Effect" or something similar.

**Questions:**

Are there any specific assumptions about the observational data and how it relates to the data with encouragement?

Can you speak some to what sort of unmeasured confounders you are assuming in your problem setup?  In Equation (1), outcome is defined as a sum of a joint function of $t$ and $x$ (treatment and measured confounders) and then an additive noise term based on $u$, whereas $t$ is a joint function of $x$ and $u$.  Why isn't $u$ part of $g_\psi$?  In the classroom participation encouragement example, can you give an example of an unmeasured confounder that might be part of $U$?

Then, in the following paragraph, you state that you "collect rich covariates $X$ as proxies for $U$".  Does this mean that you assume there are no true unmeasured confounders that represent stand-alone concepts (e.g., if variables like "socio-economic status" and "household income" weren't measured in the classroom encouragement example), only unmeasured confounders whose effects are largely captured by some variables in $X$?

On line 129, you state that traditionally, "encouragements are randomly assigned and exogenous, e.g., $U \perp \mathcal{E}$, which suggests to me that encouragements in traditional approaches may be randomly assigned based on $X$ (observed confounders) but not unobserved confounders.  You then go on to contrast this traditional assumption by saying that encouragement is often non-random, giving examples of classes and cities as potential features that could influence encouragement.  However, "classes" and "cities" seem like variables that would be easy to measure (and thus are unlikely to be unmeasured confounders).  In which case, I don't see how encouragement being non-random with respect to such features ($X \not\perp \mathcal{E}$) is at odds with the traditional assumption ($U \perp \mathcal{E}$).  Can you tell me what I'm missing here?

In Section 3.2.3 after Assumption 4, you define $CATE(T,X)$

---

### Official Review · Reviewer_GeoN · 2025-11-10

**Soundness:** 3
**Presentation:** 2
**Contribution:** 2
**Rating:** 4
**Confidence:** 3

**Summary:**

The paper introduces EnCounteR, an algorithm for estimating causal effects by combining observational data with limited encouragement (intervention) data. The method starts from a linear identification theorem that connects to classical instrumental-variable (IV) estimator and extends it to nonlinear settings through a neural network trained with an invariance-based objective. Experiments are conducted on synthetic datasets and standard causal benchmarks, claiming that EnCounteR consistently outperforms a range of existing causal estimation methods.

**Strengths:**

1. The problem setup of estimating causal effects under limited or imperfect interventions is important and practically relevant, especially in domains such where full randomization is rare.
2. The conceptual link between “encouragement environments” and IV-style identification is interesting.

**Weaknesses:**

1. **Theoretical understanding and clarity:**
   I found the theoretical sections difficult to follow. The paper introduces several assumptions abruptly (e.g., independence between \(X, U, E\)) and transitions quickly between linear, GMM, and nonlinear formulations without clearly connecting these ideas to practice.  Moreover, I am not able to assess the novelty of the formal results. The authors should clearly specify what is *new* versus what is borrowed from existing IV or GMM theory.  Since the main contributions are in theory, the authors should explicitly call out the novel aspects.

2. **Experimental design:**
   This is where I have the strongest reservations.

   **a.** All synthetic datasets are generated exactly under the structural assumptions of the proposed model (e.g., same functional form for \(f_Y\) and \(f_T\), constant covariate distributions across encouragements). This makes EnCounteR almost perfectly aligned with its own data-generating process. Competing baselines such as DeepIV, CEVAE, or AGMM are designed for different data assumptions (e.g., exogenous continuous instruments, explicit latent confounding) and are thus inherently disadvantaged.
   A more convincing evaluation would test EnCounteR on *mismatched* or partially violated settings — for example, by allowing covariate shift across environments or nonlinear confounding unrelated to \(E\).

   **b.** The nonlinear algorithm includes multiple components (encoder, adversarial discriminator, MMD loss, balancing penalty), but no ablation results are provided to show which components contribute most to performance.  An ablation table reporting results when each component is removed or weakened would significantly strengthen the empirical section.

   **c.** The nonlinear “MULT1–MULT5” synthetic benchmarks are custom-made and not publicly available. This makes it impossible to reproduce the experiments. Furthermore, the paper provides little intuition for why these benchmarks are realistic or challenging.
   Without access to code or data, the evaluation cannot be verified.

   **d.** A better use of the IHDP dataset could have been considered:
	- The IHDP benchmark provides potential outcomes for both treatments.
	- We can therefore construct a *synthetic encouragement variable* \(E\) and generate observed treatments \(T\) as a (possibly stochastic) function of \((E,X)\). The observed outcome is then \(Y = Y(T)\) using the supplied potential outcomes as-is in the original dataset.
	- This preserves the true data-generating potential outcomes (so evaluation of counterfactuals remains exact) while creating a controlled encouragement experiment.
	- Because existing baselines are already validated on IHDP, this controlled construction enables a fair, reproducible comparison in which EnCounteR can exploit explicit encouragement labels.

   **e.** The paper lacks details about hyperparameter tuning, network architectures, and optimization settings. Many baselines such as DeepIV or DeepGMM are highly sensitive to hyperparameters, and without describing the tuning procedures, the fairness of comparisons cannot be assessed.     Additionally, no code appears to be provided, which limits reproducibility.

**Questions:**

Please address the weakness in the experiments section mentioned above. Provide details for how hyperparameters were tuned for both baselines and the proposed method. Please release the code for the dataset generation as well as for the results declared in the paper to facilitate reproducibility.

---

### Note · Authors · 2025-11-16

I have read and agree with the venue's withdrawal policy on behalf of myself and my co-authors.